# A Comparative Study of Rat Urine ^1^H-NMR Metabolome Changes Presumably Arising from Isoproterenol-Induced Heart Necrosis Versus Clarithromycin-Induced QT Interval Prolongation

**DOI:** 10.3390/biology9050098

**Published:** 2020-05-13

**Authors:** Matthieu Dallons, Manon Delcourt, Corentin Schepkens, Manuel Podrecca, Jean-Marie Colet

**Affiliations:** Department of Human Biology & Toxicology, Faculty of Medicine and Pharmacy, University of Mons, Place du Parc 20, 7000 Mons, Belgium; matthieu.dallons@umons.ac.be (M.D.); manon.delcourt@umons.ac.be (M.D.); corentin.schepkens@umons.ac.be (C.S.); manuel.podrecca@umons.ac.be (M.P.)

**Keywords:** isoproterenol, clarithromycin, urine, metabonomics, ^1^H-NMR, biomarker

## Abstract

Cardiotoxicity remains a challenging concern both in drug development and in the management of various clinical situations. There are a lot of examples of drugs withdrawn from the market or stopped during clinical trials due to unpredicted cardiac adverse events. Obviously, current conventional methods for cardiotoxicity assessment suffer from a lack of predictivity and sensitivity. Therefore, there is a need for developing new tools to better identify and characterize any cardiotoxicity that can occur during the pre-clinical and clinical phases of drug development as well as after marketing in exposed patients. In this study, isoproterenol and clarithromycin were used as prototypical cardiotoxic agents in rats in order to evaluate potential biomarkers of heart toxicity at very early stages using ^1^H-NMR-based metabonomics. While isoproterenol is known to cause heart necrosis, clarithromycin may induce QT interval prolongation. Heart necrosis and QT prolongation were validated by histological analysis, serum measurement of lactate dehydrogenase/creatine phosphate kinase and QTc measurement by electrocardiogram (ECG). Urine samples were collected before and repeatedly during daily exposure to the drugs for ^1^H-NMR based-metabonomics investigations. Specific metabolic signatures, characteristic of each tested drug, were obtained from which potential predictive biomarkers for drug-induced heart necrosis and drug-induced QT prolongation were retrieved. Isoproterenol-induced heart necrosis was characterized by higher levels of taurine, creatine, glucose and by lower levels of Krebs cycle intermediates, creatinine, betaine/trimethylamine N-oxide (TMAO), dimethylamine (DMA)/sarcosine. Clarithromycin-induced QT prolongation was characterized by higher levels of creatinine, taurine, betaine/TMAO and DMA/sarcosine and by lower levels of Krebs cycle intermediates, glucose and hippurate.

## 1. Introduction

Drug-induced cardiotoxicity remains a major concern in drug development and unpredicted adverse cardiac events are too often encountered after marketing. As an illustration, 28% of drugs withdrawals were associated to cardiotoxicity [1]. Additionally, cardiac adverse effects are an important issue limiting management of patients under therapy with drugs that are still available on the market but at risk for cardiotoxicity, particularly in the field of anticancer drugs [1,2,3,4]. It is estimated that more than 20% of doxorubicin/daunorubicin/fluorouracil-treated patients will face cardiotoxicity during their treatment, so that cardiovascular disease is considered as the second cause of long-term morbidity and mortality among those treated patients who survived from their cancer [5,6]. Moreover, cardiotoxicity is not restricted to anticancer drugs but may occur during the treatment with various other drug classes [7].

Heart arrhythmias represent a complex group of electrophysiological disorders corresponding to an abnormal heart rate or rhythm. The degree of severity of the different types of arrhythmia is completely variable depending on the duration, the reversibility, the occurrence frequency, the location (atrial vs. ventricular) and the underlying mechanism. Curiously enough, arrhythmia is one of the most common cardiac issues associated with the use of various drug classes [8]. For instance, some anti-arrhythmic drugs paradoxically display pro-arrhythmic effects [9,10]. Besides arrhythmia, other heart disorders can be triggered by drugs including cardiomyocytes necrosis or apoptosis subsequently leading to heart failure. Cardiomyocytes death can be caused by several mechanisms, such as oxidative stress, mitochondrial dysfunction, heart infarct or structural impairment, among others [11,12].

Different diagnostic tools are available to evaluate the effect of drugs on the heart through the use of in silico, in vitro and in vivo studies during drug development process. In this way, an important panel of conventional methods can be used to assess cardiovascular toxicity, including biochemical assays, electrocardiogram (ECG) and heart disease diagnostic tools such as echocardiography, blood pressure measurement, histological analysis etc. ECG is considered the gold standard procedure for in vivo assessment of arrhythmia. This investigative tool informs about the heart electrical activity as an ECG waveform highlighting potential heart rhythm disorders. Drug-induced QT-interval prolongation is a kind of heart rhythm disorder, which can be classically screened by ECG. The ability of a substance to inhibit the rapid rectifier K+ channel, which usually corresponds to a pro-arrhythmic potential, can also be assessed by the in vitro human Ether-a-go-go-Related Gene (hERG) assay. Despite this large panel of assays, drug development is still facing a cardiotoxicity issue, jeopardizing the whole drug development process [13]. Obviously, more predictive and sensitive tools are needed both in drug development and for the clinical follow up of exposed patients.

Metabonomics is a systems biology predictive tool linking metabolic disruptions to pathophysiological conditions or drug exposure. Proposed in the early 2000s as a potential new investigation procedure in preclinical toxicology assessments, it has proven to be one of the most powerful predictive approach for studying organisms responses to xenobiotic exposure [14]. Basically, metabonomics investigates the metabolome by simultaneously detecting a wide range of low-molecular-weight molecules (<1500 Da) that can be found in biological matrices such as biofluids. Following metabolic changes by using NMR-spectroscopy and/or mass spectrometry (MS) should reflect early and delayed cellular responses to various stimuli [15,16,17]. ^1^H-NMR spectroscopy is a fast and non-destructive technique requiring minimal sample preparation to get spectral profiles containing a bench of signals arising from hundreds of endogenous metabolites [14,18]. Early works studied hepatic and renal toxicities in order to characterize their metabolic signatures helping the discovery of new predictive biomarkers [14,19,20]. In the past decade, some metabonomic works began to focus on drug-induced cardiotoxicity, mainly using serum and heart tissue extracts as investigative matrices [21,22]. Those works opened the way to consider metabonomics as a promising approach to discover specific biomarkers of cardiotoxicity. However, there is still work to do, particularly into the global identification of cardiotoxicity biomarkers into non-invasive matrix such as urine. Going a step further, metabonomics could be a powerful tool to help the identification of specific biomarkers profile discriminating and predicting the different kinds of cardiotoxicity-related mechanisms, (e.g., arrhythmia disorders and heart necrosis).

The objective of the present study was to characterize urine metabonomic signatures from rats receiving isoproterenol or clarithromycin to identify candidate biomarkers for heart necrosis and QT prolongation.

## 2. Materials and Methods

### 2.1. Experimental Strategy

Two independent experiments were carried out with different animal batches and manipulators. The first experiment was designed to characterize the urinary metabolome changes in a rat model of drug-induced heart necrosis. Isoproterenol (ISO) was used for modeling this kind of cardiotoxicity, as described by Yadav et al. in 2015 [23]. This necrosis model was validated by measurement of serum cardiac necrosis protein biomarkers and by histological analysis performed after the drug exposure. The second experiment was designed to highlight the urinary metabolome changes in a rat model of drug-induced QT prolongation. Clarithromycin (CLAR) was used for modeling this kind of cardiotoxicity, as described by Kmecova and Klimas in 2010 [24]. This QT prolongation model was validated by ECG monitoring during the exposure period [25,26].

### 2.2. Animals and Protocol Experiment

Ten-week-old male Wistar Han rats (Janvier Labs, Le Genest-Saint-Isle, France) were housed in our animal housing facilities according to the legislative directive 2010/63/ EU concerning the lab animal use. The rats were housed at room temperature (20–22 °C), hygrometry of 60 +/− 5% and light/dark cycle of 12 h. Seven rats received a daily subcutaneous injection of 85 mg/kg of ISO (Sigma-Aldrich, Saint-Louis, MO, USA) for two consecutive days. Six rats were daily exposed to CLAR for seven days. Pills of CLAR 500 mg (Eurogenerics, Bruxelles, Belgium) were suspended into water and rats received a dose of 100 mg/kg by gavage. To collect urine samples, rats were housed in metabolism cages with food and water ad libitum. Sodium azide 1% (1 mL/24 h fraction) was added to the urine collection tubes, which were kept at 4 °C to avoid bacterial contamination. Urine samples were collected before the exposure to the drugs and during the exposure period until the development of considered cardiac adverse effects. Samples were collected on Days 1, 1, 2, 3 and on Days 1, 1, 2, 3, 6, 7 for ISO and CLAR, respectively. Twenty-four hours after the last ISO administration, animals were euthanized by an intravenous administration of Nembutal^®^ IV and heart was surgically removed. For histological comparison purpose, three control rats were only exposed to the vehicle but underwent the entire procedure.

### 2.3. Blood Collection and Serum Biochemical Analysis

Blood collection was made before and at the end of drug exposure after euthanasia. During blood collection from the caudal vein, the rats were sedated by a ventilation of 2.5% IsoFlo^®^ at 0.5 L/min. Serum was obtained after coagulation by a 15 min. centrifugation at 3000 G. Lactate dehydrogenase (LDH) and creatine phosphate kinase (CPK) concentrations were measured in rat serum thanks to a Spotchem™ EZ SP-4430 (Arkay Europe, Amsterdam, The Netherlands) clinical analyzer.

### 2.4. Electrocardiogram and QT Interval Determination

A homemade four-leads ECG was used to determine the QT interval and the form of the T wave as arrhythmia indicators [25,26]. During the ECG measurement, animals enrolled in the CLAR study were placed in a Faraday cage and sedated by ventilation of 2.5% IsoFlo^®^ 0.5 L/min. Corrected QT interval (QTc) was manually and blindly determined according to the Bazett’s formula normalizing QT interval duration to average rat’s cardiac cycle length (RR duration) [24].

### 2.5. Histopathological Examinations

Hearts collected from rats exposed to ISO were fixed in Bouin Alcohol (picric acid 1%, ethanol 95%, formaldehyde, acetic acid and distilled water) for 48 h. Fixed tissue specimens were then dehydrated in graded ethanol bath and in butanol bath and then embedded in paraffin. Transversal paraffin sections of 5 µm thickness were cut on a Microm^®^ HM 360 (Microm Microtech, Brignais, France) microtome and mounted on glass slides. Histological examination of heart sections was carried out after staining with hematoxylin-eosin-saffron coloration.

### 2.6. Samples Preparation for ^1^H-NMR

After collection, urine samples were centrifuged at 4 °C and 1000× *g* for 5 min to remove solid wastes and were stored at −80 °C before ^1^H-NMR analysis. Then, 250 µL of phosphate buffer (0.2 M Na_2_HPO_4_/0.04 M NaH_2_PO_4_, pH 7.4) were added to 500 µL of urine to minimize pH variation. After centrifugation, 650 µL of the previous preparation were dropped into a 5 mm diameter NMR tube and 50 µL of deuterated trimethylsilylpropanoic acid (TSP) 14 mM was added as external reference.

### 2.7. Acquisition of ^1^H-NMR Urine Spectra and Data Processing

The acquisition of one dimensional ^1^H-NMR urine spectra was performed using a Bruker Avance 500 spectrometer (Bruker BioSpin GmbH, Kontich, Belgium) operating at 500.16 MHz and equipped with a 5 mm probe PABBO BB-1H. A NOESYPRESAT sequence was used for a total of 64 scans. MestReNova 11 (Mestrelab Research, Santiago de Compostela, Spain) was used for automatic baseline and phase corrections. Spectra were calibrated using the TSP peak and then binned into 0.04 ppm-width integrated regions. The data were imported into Excel (Microsoft Office^®^ 16.35) and the spectra region corresponding to water (4.5 to 5 ppm), urea (5.5 and 6 ppm), clarithromycin and isoproterenol (1.16 to 1.40 ppm) were removed before normalizing to the total area under the curve.

### 2.8. Multivariate Data Analysis, Metabolites Identification and Statistical Tests

First, the binned and integrated data were submitted to Principal Component Analysis (PCA), a non-supervised approach. Data were centered and R^2^_cum_ and Q^2^_cum_ parameters were determined. Then, data were projected to latent structure discriminant analysis (PLS-DA), a supervised method where the classes were defined for each time point of urine collection. R^2^_cum_ and Q^2^_cum_ parameters as well as *p*-value of analysis of variance of cross-validated residuals (CV-ANOVA) were determined. Variables with a variable importance in projection (VIP) value > 0.8 were selected as most discriminant variables [27]. Corresponding metabolites were identified by consulting several databases: « in house » database, Human Metabolome Database (HMDB) [28] and by the use of Chenomx Profiler software 8.3 [29].

Statistical significance of identified discriminant metabolites was determined by integrating the ^1^H-NMR peaks of each metabolite. Integrals were normalized to spectral total area. Statistical significance was determined using paired Wilcoxon test. The significance was determined at * *p*-value < 0.05, ** *p*-value < 0.01 and *** *p*-value < 0.001. From the data obtained in both studies, a heatmap was constructed from the mean values of discriminant metabolites normalized integrals, using Excel functionalities (Microsoft Office^®^ 16.35).

For non-spectral data, normality of data was determined by a Shapiro-Wilk test, considering normality with an adjusted *p*-value > 0.05 [30]. Homoscedasticity of variances were determined by a Bartlett’s test, considering homoscedasticity with a *p*-value > 0.05 [31]. For normal and homoscedastic variables, statistical significance was determined using a paired t-Test. For non-normal or heteroscedastic variables, significance was determined using paired Wilcoxon test. The significance was determined at * *p*-value < 0.05, ** *p*-value < 0.01 and *** *p*-value < 0.001.

### 2.9. Enrichment Analysis

To highlight the most relevant altered pathways for each study, a Metabolite Set Enrichment Analysis (MSEA) was performed on the discriminant metabolites using the online available software Metaboanalyst 4.0. [32]. MSEA is a powerful tool used for metabonomic data interpretation, indicating which biological pathways are possibly linked to the identified metabolites. Thus, the MSEA describes the potential biological pathways related to the metabolic signature identified through both studies. The suggested metabolic pathways are classified depending on the number of hit(s), corresponding to the number of inputted metabolites found into the identified pathways. Depending on the number of hits, an associated *p-value* is determined and can be under or above the 0.05 alpha value.

## 3. Results

### 3.1. Validation of Drug-Induced Heart Adverse Effects Specificity

As expected, a daily exposure to ISO induced heart necrosis in all exposed rats. Thus, the histological analysis revealed early stages of cell necrosis in multiple areas of myocardium from rats daily exposed to ISO for 2 days (Figure 1A), characterized by an acidosis of cardiomyocytes cytoplasm revealed by an enhanced coloring. Acidosis of cardiomyocytes cytoplasm reveals early stages of acute myocardial necrosis mechanisms in rats daily exposed to ISO for 7 days (Figure 1A). As a comparison, no sign of necrosis was found in control rats (Figure 1A). Serum levels of LDH and CPK, two enzymes released in the blood stream in case of tissues damages such as heart necrosis, were significantly increased at the end of ISO exposure, whereas no changes were found at the end of CLAR exposure (Figure 1B). Daily exposure to CLAR caused morphological alteration of the T wave, as shown on the ECG from exposed rats at Day 7 compared to those at Day 1 (Figure 1C). Similarly, the QTc was significantly prolonged in the 7-days-exposed CLAR rats as compared to the pre-exposure data (Figure 1D).

### 3.2. Urine Metabolome Changes Induced by Isoproterenol Exposure

Figure 2 shows a typical ^1^H-NMR spectrum of urine collected at Day 1 from rats exposed to ISO (Figure 2). Normalized AUC from spectral binning were analyzed by PLS-DA modeling. Scores plot highlighted three different metabolic profiles corresponding to samples before the exposure (D-1), samples from Day 1 of exposure (D1) and samples from Days 2 and 3 (D2 and D3) of exposure (Figure 3A). Scores plots suggested a dynamic change over time of the urine metabolome of ISO-exposed rats. The PLS-DA model was validated by a cross validation repeated 200 times (Figure 3B). Loadings plot indicated the variables responsible for the metabolic profile separation, with their corresponding metabolite assignments (Figure 3C). All discriminant metabolites (VIP score > 0.8) are described in Table 1. A daily exposure to ISO was characterized by decreases in citrate, α-ketoglutarate, succinate, hippurate and dimethylamine (DMA) or sarcosine concentrations and an increased glucosuria. Increases in taurine and betaine or trimethylamine N-oxide (TMAO) were seen on the first day of exposure. Higher creatinuria and creatininuria were noticed on Days 2 and 3.

All identified discriminant metabolites were then inputted into the MetaboAnalyst 4.0 online software for a Metabolite Set Enrichment Analysis (MSEA). This analysis allowed inferences in the most likely metabolic pathways suggested by ^1^H-NMR-based metabonomic data (Figure 4). Glucose-alanine metabolism, Krebs cycle, carnitine synthesis, acetyl groups transfer in mitochondria, fatty acids oxidation, taurine metabolism, several amino acids metabolisms (glycine, serine, arginine, proline, methionine and glutamate) as well as the « Warburg effect » were suggested to be mostly involved in the dynamic metabolic signature of ISO-exposed rats.

### 3.3. Urine Metabolome Changes Induced by Acute Clarithromycin Exposure

Figure 2 shows typical ^1^H-NMR spectrum of urine from rats exposed to CLAR on Day 3 (Figure 2C). Normalized AUC from spectral binning were analyzed by PLS-DA modeling. Scores plot allowed the discrimination between the samples collected on Day 3 and samples collected at any other time point (Figure 5A), suggesting that the metabolic profiles observed on Days 1, 2, 6 and 7 are closer to the pre-exposure baseline. The scores plot highlighted a dynamic change over time of the urine metabolome of the CLAR-exposed rats. This metabolic change was characterized by a recovery towards the pre-exposure baseline at the end of the study. The PLS-DA model was validated by a cross validation repeated 200 times when samples collected on Days −1, 1, 2, 6 and 7 were considered as a single class (Figure 5B). Loadings plot indicated the variables responsible for the metabolic profile separation, with their corresponding metabolite assignments (Figure 5C). All discriminant metabolites (VIP score > 0.8) are listed in Table 2. The metabolic profile obtained on Day 3 was described by increases in citrate, α-ketoglutarate, succinate and hippurate concentrations and by decreased levels of creatinine, taurine, betaine/TMAO and DMAO/sarcosine. Although the multivariate analysis identified similar profiles on Days −1, 1, 2, 6 and 7, a more accurate analysis of the spectra enabled the identification of significant changes between Days −1 and Days 1–2, and between Day −1 and Days 6–7. Thus, citrate, α-ketoglutarate, succinate, hippurate and betaine or TMAO levels were reduced on Days 1 and 2, while creatinine, DMA/sarcosine and taurine levels were enhanced. On Days 6 and 7, decreases of citrate, α-ketoglutarate, succinate and hippurate levels and increases of betaine/TMAO, creatinine, DMA or sarcosine and taurine levels were observed.

All identified discriminant metabolites were next inputted into the MetaboAnalyst 4.0 online software for a Metabolite Set Enrichment Analysis (MSEA). This analysis suggested the most likely metabolic pathways based on the ^1^H-NMR-based metabonomic data (Figure 6). Krebs cycle, carnitine synthesis, fatty acids oxidation, malate-aspartate shuttle and several amino acids metabolisms (Glycine, serine, methionine, glutamate, arginine, proline, alanine), as well as the « Warburg effect, » were proposed to be highly involved in the dynamic metabolic signature of CLAR-exposed rats.

### 3.4. Multivariate Analysis of Both Experiments Combination

A PLS-DA was performed with metabonomic data from both ISO and CLAR exposure experiments. The scores plot indicates different metabolic profiles according to the drug administration (Figure 7A). Pre-exposed samples from both ISO and CLAR experiments were gathered in the same frame, suggesting their close similarity. A distinct metabolic profile was obtained for all ISO exposure samples from Day 1 to Day 3. A third metabolic profile was determined for the samples from the CLAR-exposed group on Day 3. In this model, data from Days 1, 2, 6 and 7 of CLAR exposure were excluded because of their similarities with pre-exposure data. The PLS-DA model was validated by a cross validation repeated 200 times (Figure 7B). Next, a heatmap was generated based on the ^1^H-NMR AUC mean values of the identified discriminant metabolites, all time points for both ISO and CLAR exposure being included (Figure 7C). Both drug-exposures were characterized by a decrease in hippurate and Krebs cycle intermediates (citrate, α-ketoglutarate and succinate) at all considered time points. However, this decrease was more pronounced in CLAR-exposed rats. Major drug-related metabolic differences included variations in creatine, creatinine and DMA or sarcosine levels. Creatine levels progressively increased during the ISO exposure, while no change was found in CLAR exposure. Creatinine levels were highly increased in CLAR samples, while minor changes were highlighted in ISO samples. DMA/sarcosine concentration was decreased during ISO exposure while it increased during CLAR exposure. Glucose concentration was enhanced in ISO exposure samples and reduced on Day 3 during CLAR exposure. Taurine also fluctuated differently according to the given drug. Indeed, taurine levels were slightly increased and maintained during the whole CLAR exposure, whereas the ISO exposure induced an important increase of taurine at the beginning of the ISO exposure, followed by a recovery to pre-test and control values. Betaine/TMAO levels were drastically reduced on Day 1 of ISO exposure before recovery of baseline levels, while a slight increase was detected on Day 7 of CLAR exposure.

## 4. Discussion

Approximately 30% of drug withdrawals are due to unanticipated cardiac adverse events [33]. Nowadays, drug-induced cardiotoxicity remains a major health issue still associated to a poor understanding and predictivity of underlying cellular toxicity mechanisms. Cardiomyocytes are highly differentiated cells whose function and viability can be deeply impaired by exposure to any chemical compromising intracellular ions equilibrium, energetical homeostasis or pro-survival cellular mechanisms [34]. Mainly related to voltage gated ion channels disturbances, arrhythmia is one of the main disorders associated with the use of cardiotoxic drugs [35]. In addition, some cardiotoxic drugs cause cellular dysfunctions overwhelming survival mechanisms and leading to irreversible cardiomyocytes death. Nowadays, such damaging events, known as drug-induced heart failure and cardiac necrosis, are limiting factors in the management of too many patients, especially during chemotherapy. Animal models have been extensively used to explore in-depth cellular mechanisms involved in the onset of cardiotoxicity. To this end, two prototypical cardiotoxic drugs, isoproterenol and clarithromycin, were acutely administrated to rats to assess, respectively, two well-known cardiac dysfunctions: QT-prolongation and cardiomyocytes death associated to necrosis. Nowadays, conventional assessment methods such as ECG or serological analyses, commonly used to predict those cardiotoxic events, reflect entrenched cardiomyocytes dysfunctions substantiating their lack of predictivity and sensitivity. As a matter of fact, there is an obvious need for developing new tools to ensure a better prediction of drug-induced heart adverse effects in all phases of the drug development including post-marketing. In this respect, our study was designed to highlight potential new predictive biomarkers of early and acute heart adverse effects that could be accessible with a non-invasive procedure, using ^1^H-NMR metabonomic investigation of urine samples. Data acquired in both metabonomic studies were initially analyzed separately from each other, in order to first identify urinary metabolic changes that could specifically witness each type of cardiotoxicity. Secondly, all data were analyzed together in a global multivariate data analysis in order to appreciate the ability of ^1^H-NMR-based metabonomics to discriminate two different mechanisms of drug-induced cardiotoxicity.

Cardiac cell necrosis induction in ISO-exposed rats was confirmed both histologically and by increased *serum* levels of LDH and CPK, two conventional protein markers of heart necrosis [36]. The metabonomic model of ISO-induced heart necrosis was based on a PLS-DA performed on urine samples collected from ISO-exposed rats. This model highlighted multiple metabolic changes, which were gathered in distinct functional clusters and discussed hereafter. A first metabolic cluster was indicative of a significant alteration in heart energy synthesis. Under normal conditions, 70% to 90% of heart ATP is produced from the mitochondrial fatty acids oxidation that provides acetyl-CoA to Krebs cycle, which is the main metabolic energy pathway [37]. The urine levels of several TCA intermediates (citrate, α-ketoglutarate, succinate) were progressively decreased during the exposure to ISO. The concept of metabonomics augurs that any variation in the intracellular concentration of those metabolites capable of crossing the cell membrane will be spontaneously and proportionally reflected in the extracellular compartment due to homeostasis. Therefore, during the ISO-exposure, a progressive loss of TCA intermediates in urine most likely reflects a lowest mitochondrial activity in cells exposed to the drug. This altered TCA cycle activity could be linked to an ongoing heart necrosis process or even organ failure [38,39,40]. Li et al. also highlighted metabolic changes in *serum* of rats exposed to ISO suggesting an early alteration of heart mitochondrial energy metabolism, using UPLC-Q-TOF-MS [22]. Some evidence suggests that the cardiotoxicity of many drugs is related, at least partially, to a mitochondrial deficiency. Indeed, exposure to recognized cardiotoxic drugs were characterized by TCA dysfunction [12]. Additionally, Asnani et al. demonstrated a correlation between citrate level in plasma and the severity of doxorubicin-induced cardiotoxicity in patients with breast cancer [41].

Moreover, a higher glycosuria was observed in our study and could suggest a lower tissue consumption, reflecting the mitochondrial energy alteration and a decreased glycolytic activity. Changes in glucose consumption are likely to be associated with various heart diseases. A higher serum level of glucose was reported in acute myocardial infarction, using ^1^H-NMR metabonomics [42]. Authors observed that the higher glycemia was associated with an early elevation of ketone bodies levels in the serum of patients suffering from myocardial infarction. Their results concurred with those of Laborde et al. in 2014 [43]. Both studies suggest that hypoxia “mimics” the physiological changes that occur in diabetes. The low energy yield of glucose metabolism forces cells to metabolize fatty acids for ATP production and ketone bodies release. Increased glucose levels in serum and/or heart tissue were also highlighted in rats suffering from heart acute toxicity induced by Venenum Bufonis [44] and in mice suffering from cardiotoxicity induced by doxorubicin [45]. However, the observed higher glycosuria and other changes related to energy metabolism could also reflect an insulin resistance due to ISO exposure. Hoff and Koh recently reported a case of ISO-induced diabetic ketoacidosis in a 77-year-old female patient treated with ISO for atrioventricular block [46]. In addition, several evidences suggest that ISO exposure could disturb the insulin signaling pathway [47,48,49]. Further investigations should be pursued to specify the role of insulin resistance in the metabonomic signature we reported.

The metabonomics results described a gradual enhancement of creatine level, while creatininuria progressively and concomitantly decreased. Creatine is biosynthesized from amino acids in liver, pancreas and kidney and plays an important role in muscles and heart as energy storage system. Creatinine is the end-product of creatine and phosphocreatine, and is released in blood compartment in order to be eliminated in urine [50]. The increased concentration of creatine in urine may be due to its higher outflow in blood because of the rupture of plasma membrane in necrotic cardiomyocytes [51]. Moreover, the larger creatine/creatinine ratio suggests a lower creatine degradation in suffering cardiomyocytes.

The metabonomic analysis of urine during ISO exposure showed decreased levels of betaine or TMAO and DMAO or sarcosine, all cell osmolytes that are specifically incorporated or released from cells to counteract any osmotic stress. The lower energy level due to ISO exposure and appreciated by the metabolic indicators of a mitochondrial alteration is suspected to consecutively cause an osmotic stress in cardiomyocytes. As a compensatory mechanism, cells could incorporate those osmolytes to preserve their volume and integrity [52]. This incorporation is reflected by a lower elimination of these compounds in urine.

Taurine is an abundant metabolite in the heart where it is known to play several functions that help maintain both cellular osmosis and integrity. Although taurine is known to be involved in heart osmoregulation too, its urine excretion evolves in the opposite direction compared to the other osmolytes. Several authors reported on a functional role of taurine for cell membrane stabilization in heart [53,54,55]. Therefore, the enhancement of taurine levels in urine could reflect damage to cardiomyocytes membranes, as a hallmark of necrosis. Significant alterations in serum lipid content, underlying cell membrane damages due to accelerated phospholipids degradation, were previously described in rats and dogs exposed to ISO [56,57]. In our study, the higher taurine level in urine occurred mainly on Day 1 of the ISO exposure, suggesting that it could be a very early marker of cardiomyocytes membrane alteration, before the release of cytosolic contents such as creatine most likely due to a later event in cell necrosis. Moreover, taurine is suspected to have protective effect against ISO-induced heart damages. Ohta et al. observed a partial protective effect of an oral intake of taurine on ISO-exposed chick heart [58]. However, as taurine seems to have other important physiological functions in the heart, such as antioxidant properties, cardiac ionic modulation and sodium and calcium homeostasis [59], the early changes in taurine level could also reflect some adaptation to high cardiomyocytes stress. Therefore, precocious higher taurinuria could be considered as a potential cellular-stress response signal rapidly triggered to cope with heart necrosis onset. Such modulations in taurine levels were also highlighted in other cardiotoxicity studies, supporting the potential link between cardiotoxicity and changes in taurine level and metabolism. For example, Jensen et al. reported an alteration in taurine metabolism in plasma and heart tissue from mice exposed to sorafenib [60], while Yoon et al. observed an increase level of taurine in human cardiomyocytes AC16 cells exposed to spinochrome D, used as a protective agent against doxorubicin-induced cardiotoxicity [61]. In a previous study, we highlighted an increased intracellular taurine concentration in H9C2 cells exposed to doxorubicin after 24 h [62]. Finally, Gramatyka et al. found an alteration in taurine metabolism after exposure of human cardiomyocytes to ionizing radiation.

The QTc prolongation in CLAR-exposed rats was confirmed by ECG measurement of QTc interval. The absence of a simultaneous heart necrosis was verified by unchanged serum levels of LDH and CPK over the exposure time-window. The metabonomic model of CLAR-induced QTc prolongation was based on a PLS-DA performed on urine samples collected from CLAR-exposed rats. This model highlighted multiple metabolic changes which were gathered in functional clusters and discussed hereafter.

The urine concentrations of several TCA intermediates (citrate, α-ketoglutarate, succinate) were decreased during the exposure to CLAR, reaching their lowest level at D3. According to the concept of metabonomics, such urine changes most likely reflect a lowest mitochondrial activity in affected cardiomyocytes that could predict arrhythmia observed in D7. Indeed, several authors suggested that a loss of mitochondrial function is a key contributor of arrhythmia [63,64], probably by a modulation of the redox or energy signaling pathways involved in the ion channels and transporters regulation [65,66,67,68]. It was reported that CLAR can cause hypoglycemia, particularly in diabetic patients or in case of drug combination [69,70,71]. The decreased level of glucose in urine on Day 3 of CLAR exposure may reflect a lower glycaemia that could jeopardize the glucose availability for heart energy metabolism, and thus, accentuate the depletion of heart energy production. On the opposite, the lower glycosuria may also suggest a decreased urinary excretion of this metabolite, and therefore, a higher glycaemia, which could be correlated with QTc prolongation. An explorative metabonomic study of QTc interval associated patterns in shift workers by Campagna et al. in 2016 pointed out serum metabolic changes linked to glycemic homeostasis, suggesting a causative relationship between a deregulation of blood glucose level and QTc prolongation. The authors identified a positive correlation between glycaemia, serum glucose concentration and QTc values [72]. In our study, glycaemia was not measured. Therefore, these opposite hypothetical mechanisms about the glucose fate should enter in consideration for further investigation of CLAR-induced QT prolongation mechanisms.

The results highlighted a higher creatininuria during the whole duration of CLAR exposure. The degradation of creatine into creatinine is essentially a non-enzymatic process that is regulated by complex mechanisms such as allosteric regulation, chemical modification or changes in the membrane permeability [50]. CLAR could promote the catabolism of creatine into creatinine, as reflected by its enhanced urine excretion and/or enhance membrane permeability to creatinine. Because of the implication of creatine as local energy storage in heart, this metabolic alteration may also be linked to the lower energy production pointed out by observed changes in Krebs cycle intermediate levels. Further investigations are needed to better understand how CLAR could jeopardize creatine metabolism and its link with arrhythmia.

The metabonomic findings pointed out a decrease of hippurate level. This decrease may be explained by an increase in glutathione (GSH) synthesis. Indeed, glycine which is one of the three GSH precursors is also used for hippurate synthesis. A higher level of GSH is a cell adaptive mechanism to counteract oxidative stress which is a hallmark of mitochondrial dysfunction [73].

Finally, an enhancement of several osmolytes levels in urine of CLAR-exposed rats (betaine or TMAO, taurine, DMA or sarcosine) was also noticed. Cardiac arrhythmia involves a perturbation of ion channels function unsettling cardiomyocytes osmolality. Therefore, the changes of osmolytes levels could be an adaptive mechanism triggered to maintain the cell volume homeostasis.

Besides promoting harmful QT prolongation, CLAR is also responsible for other toxicities. During Phases II and III of CLAR clinical trial involving 3437 patients some adverse events were reported in 20% of the 3437 patients, and only 1% of these were considered as severe. Most of those adverse events (11% of patients) were digestive system upsets [74]. No significant hematological, hepatic or renal toxicities were reported. After marketing, several cases of CLAR-induced neurotoxicity in adults were reported [75]. Considering those multiorgan CLAR-induced adverse effects, it would be unwise to associate metabolic changes with cardiotoxicity alone.

A PLS-DA model gathering the data from both cardiotoxicity studies was next performed. This model was able to discriminate urine metabolic profiles from individuals exposed to ISO from those exposed to CLAR. Metabolic changes were specifically induced by each drug, paving the way for the development of potential biomarkers able to discriminate drug-induced heart necrosis and drug-induced QT prolongation. The metabolic profile of ISO-induced heart necrosis mainly indicates an alteration of cardiomyocytes membrane integrity, as evidenced by the release of compounds outside the cells (early excretion of taurine and progressive excretion of creatine associated with decreasing creatinine production) and cellular uptake of osmolytes (decreased excretion of betaine/TMAO, DMA/sarcosine). The metabolic profile of CLAR-induced QT prolongation mainly consisted in functional alterations of ion channels and biological membranes (higher excretion of creatinine), oxidative stress with the onset of some protective mechanisms (decreased excretion of hippurate) and cell release of some osmolytes (increased excretion of betaine/TMAO, DMA/sarcosine, taurine). Common features were found in both cardiotoxicities. Indeed, both were characterized by mitochondrial energy metabolism alteration (decreased excretion of citric acid cycle intermediates), suggesting that mitochondrial impairment might be a common hallmark of various cardiotoxicities or tissue impairments. The only different metabolic change related to energy metabolism was the fate of glucose, which was more excreted in ISO-exposed rats and less excreted in CLAR-exposed rats. This opposite fate could reflect distinct effect on glucose metabolism and uptake, as well as other physiological effects induced by those drugs on other organs and tissues.

Applied to toxicology, the metabonomic investigation from biofluid samples such as urine or serum reflects the global metabolic change occurring in a complex organism during an exposure to a xenobiotic. Therefore, the metabolic variations observed in urine may be caused by simultaneous actions of the xenobiotic on different organs and tissues. The main limitation of this global metabolic approach is the discernment of metabolite changes induced by cardiac adverse effects from those induced by xenobiotic interactions with other organs/tissues. As an illustration, a decrease of urine level of hippurate might be associated with hepatotoxicities as well as nephrotoxicities [76,77,78,79]. The hope with this global approach is the identification of highly specific signatures of drug-induced heart necrosis and drug-induced QT prolongation. For this purpose, our study lays interesting biological foundations for the development of a predictive model of drug-induced cardiotoxicity, provided that additional cardiotoxic agents are added to the database to refine the identification of common specific biomarkers for each type of cardiotoxicity.

## 5. Conclusions

Our metabonomic investigation on rat urine samples allowed to point out some potential metabolic biomarkers discriminating drug-induced heart necrosis from drug-induced QT prolongation that could enter into consideration for developing new early preclinical and clinical biomarkers of cardiotoxicities accessible from a non-invasive sample collection. However, further investigations are required to validate their biological correlations and confirm their cardiac origin. The next challenge will be the development of a predictive expert system by including data from other drugs causing heart necrosis and QT prolongation to refine the metabolic signature by identifying the common metabolic changes.

## Figures and Tables

**Figure 1 biology-09-00098-f001:**
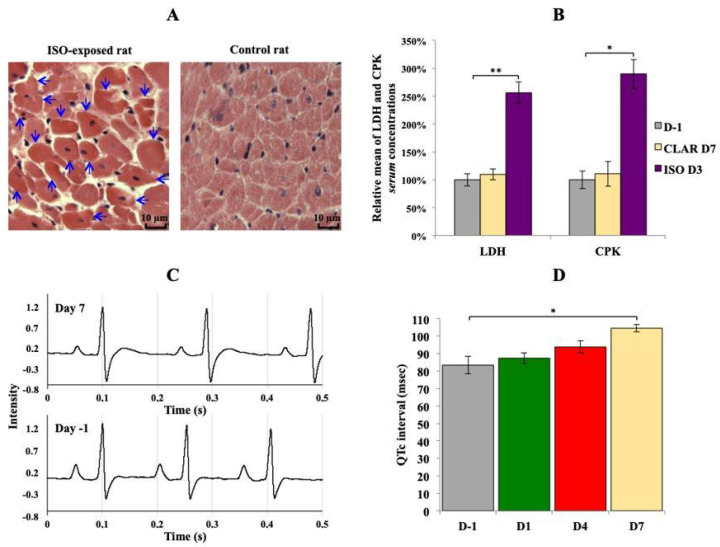
Validation of the considered cardiotoxicological mechanism for each experiment. (**A**) Transversal slide of heart from rat daily exposed to isoproterenol (ISO) at the end of the exposure period (3 days) and from control rat. Arrows indicate early stage of cardiomyocytes necrosis. Lens 250×. (**B**) Relative means ± SEM of lactate dehydrogenase (LDH) and creatine phosphate kinase (CPK) *serum* concentrations from rats before the exposures and at the end of ISO and clarithromycin (CLAR) exposures. Paired t-Test: * *p*-value < 0.05; ** *p*-value < 0.01. (**C**): Rat electrocardiogram (ECG) before the daily CLAR exposure and at the end of the exposure period (7 days). (**D**) Means ± SEM of corrected QT interval (QTc) measured on ECG from rats before the daily CLAR exposure and at days 1, 4 and 7 of the exposure period. Paired t-test: * *p*-value < 0.05.

**Figure 2 biology-09-00098-f002:**
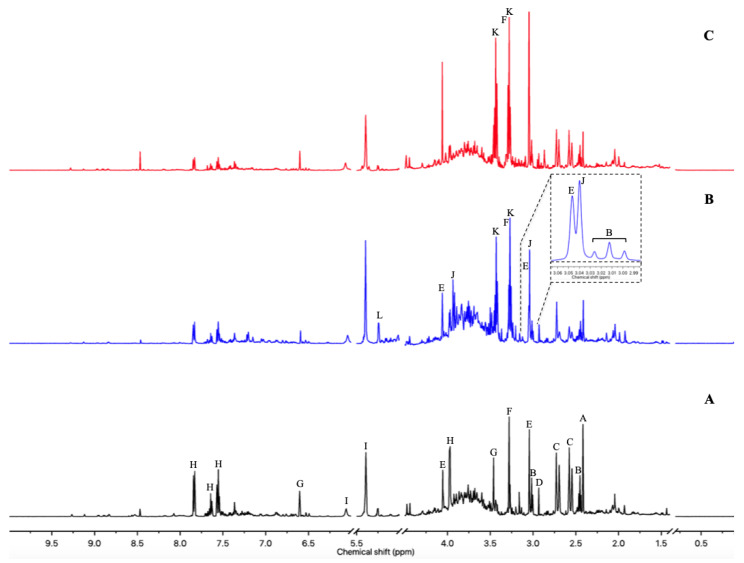
500 MHz ^1^H-NMR urine spectra from Wistar Han rats. (**A**) One day before any exposure. (**B**) First day of the ISO exposure. (**C)** Third day of the CLAR exposure. Metabolite assignments: A—Succinate; B—α-Ketoglutarate; C—Citrate; D—Dimethylamine (DMA) or Sarcosine; E—Creatinine; F—Betaine or Trimethylamine N-oxide (TMAO); G—trans-Aconitate; H—Hippurate; I—Allantoin; J—Creatine; K—Taurine; L—Glucose.

**Figure 3 biology-09-00098-f003:**
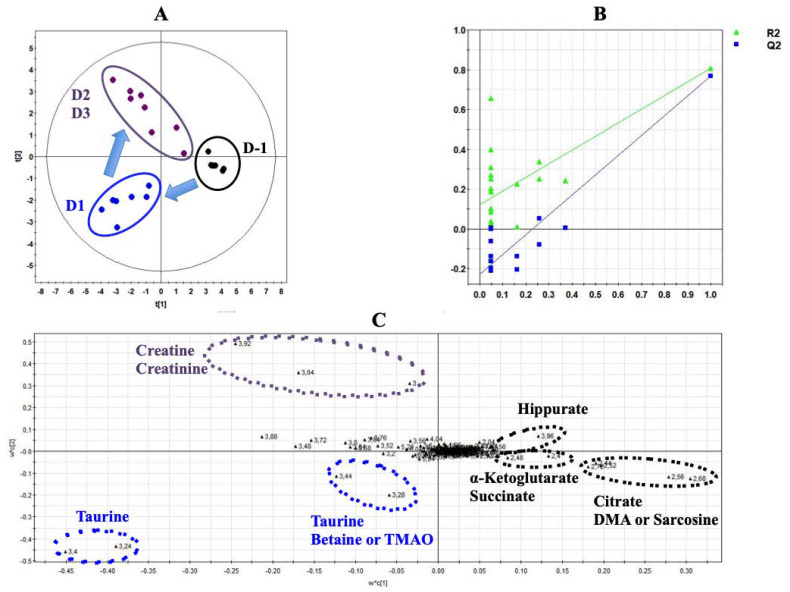
Projection to latent structure discriminant analysis (PLS-DA) modeling of metabonomic study performed on urine samples from rats daily exposed to ISO. (**A**) Scores plot from ^1^H-NMR spectra of rat urine samples at different time points: One day before the exposure (•) vs. day 1 of the exposure period (•) vs. days 2 and 3 of the exposure period (•). R^2^_cum_ = 0.84; Q^2^_cum_ = 0.8; Hotelling’s T2 = 0.95; *p*-value of analysis of variance of cross-validated residuals (CV-ANOVA) < 0.001. Arrows indicate the direction of the metabolic changes. (**B**) Cross-validation plot (R^2^ in green, Q^2^ in blue) with a permutation test repeated 200 times. The Y axis intercepts were R^2^ = (0.0; 0.118) and Q^2^ = (0.0; −0.22). (**C**) Loadings plot from ^1^H-NMR spectra of rat urine samples at different time points with corresponding identified metabolites.

**Figure 4 biology-09-00098-f004:**
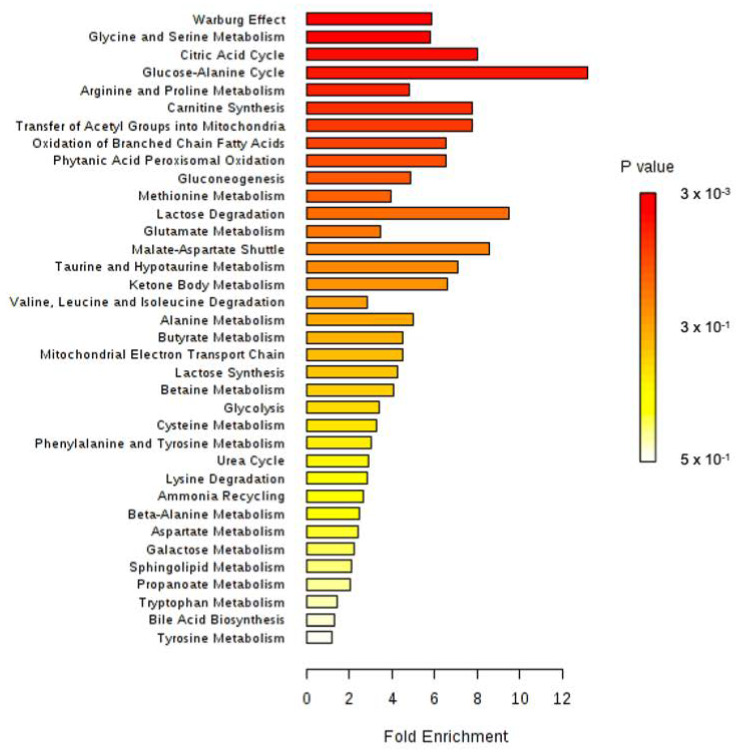
Metabolite Set Enrichment Analysis performed on ^1^H-NMR metabonomic data of ISO exposure, using MetaboAnalyst 4.0 online software. 3 × 10^3^.

**Figure 5 biology-09-00098-f005:**
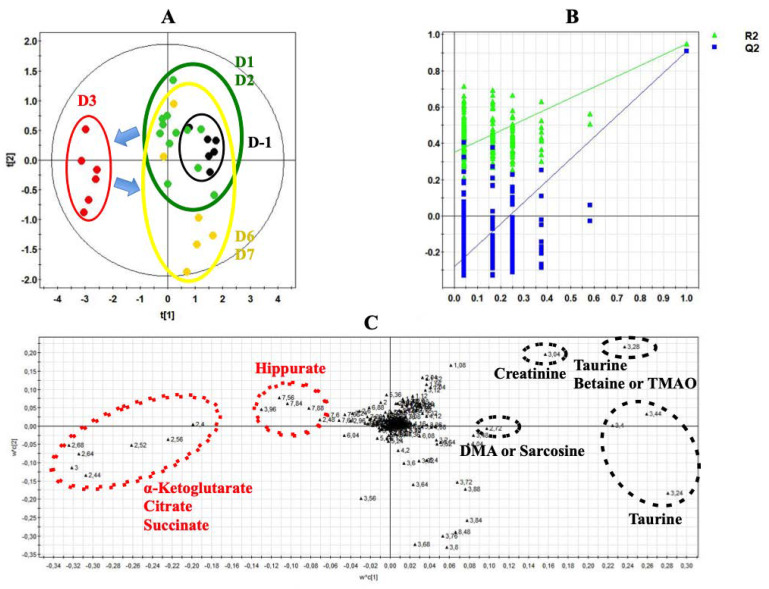
PLS-DA modeling of metabonomic study performed on urine samples from rats daily exposed to CLAR. (**A**) Scores plot from ^1^H-NMR spectra of rat urine samples at different time points One day before the exposure (•) vs. days 1 and 2 of the exposure period (•) vs. day 3 of the exposure period (•) vs. days 6 and 7 of the exposure period (•). R^2^_cum_ = 0.65; Q^2^_cum_ = 0.44; Hotelling’s T2 = 0.95; *p*-value (CV-ANOVA) < 0.01. Arrows indicate the direction of the metabolic changes. (**B**) Cross-validation plot if days -1, 1, 2, 6 and 7 are grouped in the same class (R^2^ in green, Q^2^ in blue) with a permutation test repeated 200 times. The Y axis intercepts were R^2^ = (0.0; 0.356) and Q^2^ = (0.0; −0.291). (**C**) Loadings plot from ^1^H-NMR spectra of rat urine samples at different time points with corresponding identified metabolites.

**Figure 6 biology-09-00098-f006:**
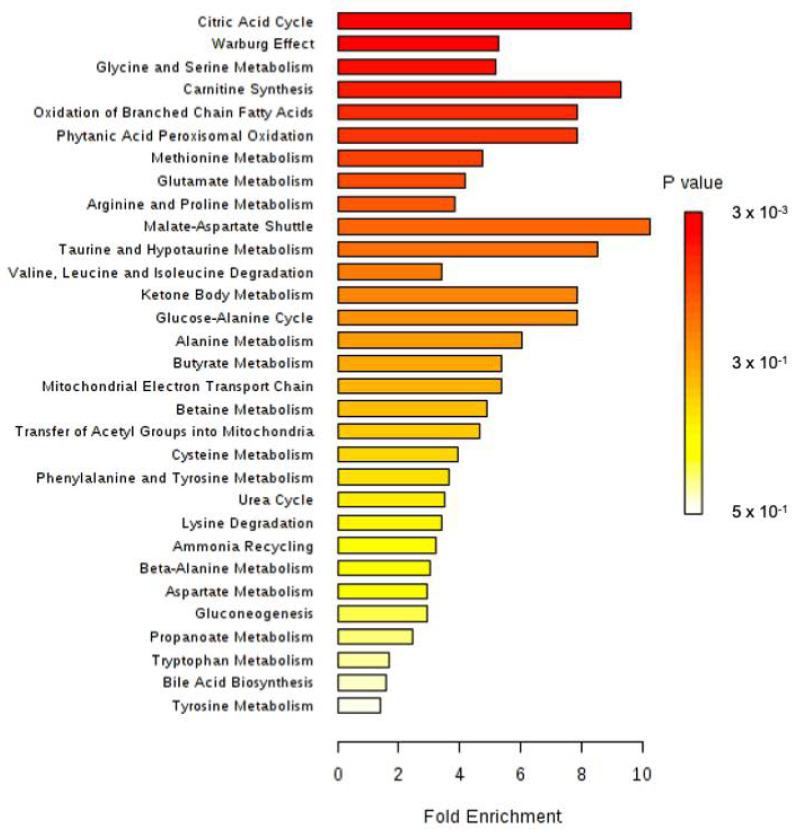
Metabolite Set Enrichment Analysis performed on ^1^H-NMR metabonomic data of CLAR exposure, using MetaboAnalyst 4.0 online software.

**Figure 7 biology-09-00098-f007:**
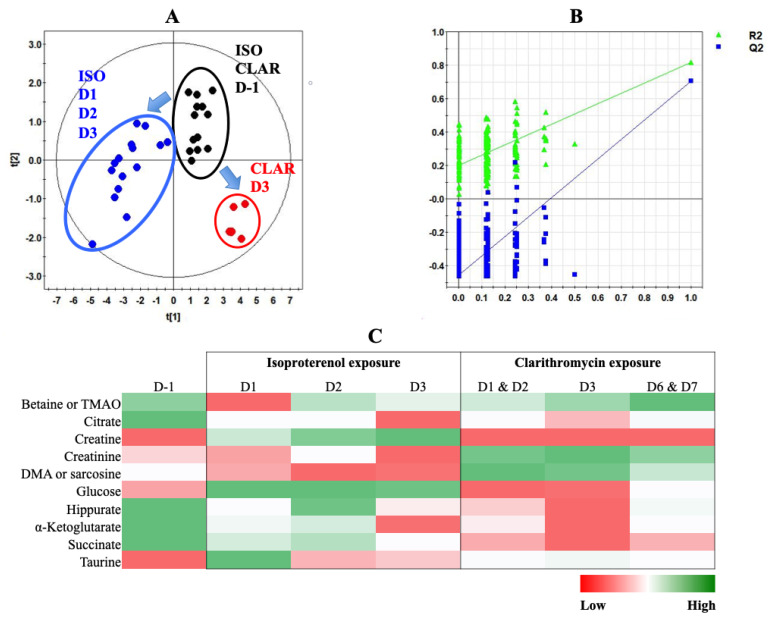
PLS-DA modeling of metabonomic study performed on urine samples from rats daily exposed to ISO or CLAR. (**A**) Scores plot from ^1^H-NMR spectra of rat urine samples at different time points: one day before the exposure (•) vs. days 1, 2 and 3 of the ISO exposure period (•) vs. day 3 of the CLAR exposure period (•). R^2^_cum_ = 0.88; Q^2^_cum_ = 0.81; Hotelling’s T2 = 0.95; *p-value* (CV-ANOVA) < 0.001. Arrows indicate the direction of the metabolic changes. (**B**) Cross-validation plot (R^2^ in green, Q^2^ in blue) with a permutation test repeated 200 times. The Y axis intercepts were R^2^ = (0.0; 0.2) and Q^2^ = (0.0; −0.467). (**C**) Heatmap constructed from relative means of discriminant metabolites normalized integrals for ISO and CLAR exposures.

**Table 1 biology-09-00098-t001:** Identified discriminant metabolites with corresponding chemical shifts for the ISO study.

Metabolites (VIP)	Chemical Shifts	Day 1	Days 2 and 3
Betaine or TMAO (0.86)	3.28 (s)	↓ *	↓
Citrate (4.44)	2.56 (d) 2.68 (d)	↓ *	↓ *
Creatine (2.46)	3.04 (s) 3.92 (s)	↑ *	↑ *
Creatinine (2.46)	3.05 (s) 4.05 (s)	↓	↓
DMA or sarcosine (2.64)	2.72 (s)	↓	↓
Glucose (1.63)	3.23 (dd) 3.39 (m) 3.45 (m) 3.52 (dd) 3.72 (m) 3.82 (m) 3.88 (dd) 5.22 (d)	↑ *	↑ *
Hippurate (1.76)	3.96 (d) 7.54 (m) 7.62 (m) 7.83 (dd)	↓	↓
α-Ketoglutarate (2.78)	2.44 (t) 3.00 (t)	↓ *	↓ *
Succinate (1.95)	2.40 (s)	↓ *	↓ *
Taurine (6.56)	3.24 (t) 3.40 (t)	↑ *	↑
Unknown (3.10)	3.88	↑	↑
Unknown (2.24)	3.72	↑	↑
Unknown (1.54)	3.52 3.64 3.68 3.76 3.80 3.84	↑	↑

Peaks multiplicity is indicated between brackets (s = singulet, d = doublet, t = triplet, m = multiplet). Arrows indicate metabolites concentration changes compared to pre-exposure samples: ↑ means an increased level, ↓ means a decreased level. Variable importance in projection (VIP) values are specified between brackets. Paired Wilcoxon test: * *p*-value < 0.05.

**Table 2 biology-09-00098-t002:** Identified discriminant metabolites with corresponding chemical shifts for the CLAR study.

Metabolites (VIP)	Chemical Shifts	Days 1 and 2	Day 3	Days 6 and 7
Betaine or TMAO (3.42)	3.28 (s)	↓	↓	↑
Citrate (4.72)	2.56 (d) 2.68 (d)	↓ **	↓	↓ **
Creatinine (2.35)	3.05 (s) 4.05 (s)	↑ **	↑	↑ **
DMA or sarcosine (1.62)	2.72 (s)	↑ *	↑	↑
Hippurate (1.71)	3.96 (d) 7.54 (m) 7.62 (m) 7.83 (dd)	↓ *	↓	↓ **
α-Ketoglutarate (4.80)	2.44 (t) 3.00 (t)	↓ **	↓	↓ **
Succinate (2.98)	2.40 (s)	↓ **	↓	↓ **
Taurine (3.81)	3.24 (t) 3.40 (t)	↑ **	↑	↑ **
Unknown (1.07)	2.48	↓	↑	↓
Unknown (4.79)	2.64	↓	↓	↑
Unknown (1.12)	3.72 3.76 3.84 3.88	↓	↓	↑
Unknown (0.92)	8.48	↓	↓	↑

Peaks multiplicity is indicated between brackets (s = singulet, d = doublet, t = triplet, m = multiplet). Arrows indicate metabolites concentration changes compared to pre-exposure samples. VIP values are specified between brackets. Paired Wilcoxon test: * *p* value < 0.05, ** *p* value < 0.01.

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
