# Peer review of "A Comparative Study of Rat Urine 1H-NMR Metabolome Changes Presumably Arising from Isoproterenol-Induced Heart Necrosis Versus Clarithromycin-Induced QT Interval Prolongation"

_biology, 2020, doi:10.3390/biology9050098_

Round 1
Reviewer 1 Report
I think is a good and interesting manuscript.
I'd like suggest only some modifications in order to improve current version.
Summing up:
--POINTs of STRENGHT:
a) idea/novelty
b) well established argumentation/discussion of the study
c) methods explaination are detailed and especially I have appreciated very good statistical choices
d) high quality of inserted figures
--POINT of WEAKNESS:
e) English language must be partially checked (i.e. minor spelling errors such as 'characterisic' at line 28 or not totally clear sentences like 'One no longer counts drugs withdrawn from the market or stopped during clinical trials due to unpredicted cardiac adverse events' at lines 15-16 rather than 'As an illustration, 28% of drugswithdrawals were associated to cardiotoxicity' at lines 38-39)
f) moreover, at line 61 probably, it might sound better the form 'etc.' instead of '...'
g) lines 104-105: Please provide, if possible a valid reference, also for ECG monitoring in any similar setting.
h) lines 131-132: About Bazett's calculation -> Was it automatic or not? And if it was performed by any observer what about intra- and inter observer variability?
i) ABOUT REFERENCES. Please re-check if all are in accordance with authors' guidelines (i.e. number 55 --> Free Radical Biology and Medicine is a wrong abbreviation)
Good work and best regards,
LG
Author Response
We would like to acknowledge the Reviewer 1 for his/her general opinion. Herewith our answers to the Reviewer’s comments:
[POINT 1] English language must be partially checked (i.e. minor spelling errors such as 'characterisic' at line 28 or not totally clear sentences like 'One no longer counts drugs withdrawn from the market or stopped during clinical trials due to unpredicted cardiac adverse events' at lines 15-16 rather than 'As an illustration, 28% of drugs withdrawals were associated to cardiotoxicity' at lines 38-39)
[RESPONSE 1] We checked the English language and made corrections in consideration to the Reviewer’s detected minor spelling errors through all the manuscript. We replaced the sentence “One on longer counts drugs withdrawn from the market or stopped during clinical trials due to unpredicted cardiac adverse events” by “There are a lot of examples of drugs withdrawn from the market or stopped during clinical trials due to unpredicted cardiac adverse events” (lines 15-16) to be in accordance with the idea of the sentence “'As an illustration, 28% of drugs withdrawals were associated to cardiotoxicity” (lines 38-39). Moreover, we changed the sentence “Isoproterenol (ISO) was used for this purpose” (line 99) by “Isoproterenol (ISO) was used for modeling this kind of cardiotoxicity” to make the idea clearer. With the same objective, we changed the sentence “Clarithromycin (CLAR) was used for this purpose” (lines 103-104) by “Clarithromycin (CLAR) was used for modeling this kind of cardiotoxicity”.
[POINT 2] moreover, at line 61 probably, it might sound better the form 'etc.' instead of '...'
[RESPONSE 2] “…” were replaced by “etc.” (line 61)
[POINT 3] lines 104-105: Please provide, if possible a valid reference, also for ECG monitoring in any similar setting.
[RESPONSE 3] Kmecova and Klimas, 2010, (referenced in the main text) previously reported a model of clarithromycin exposure to investigate the heart rate correction of QT duration in rats. Our clarithromycin exposure conditions (dose, frequency and duration) were highly inspired by the exposure conditions described in this study. Their experimental conditions were known to induce a QT prolongation in rats, essential for metabonomics signature investigation of this mechanism. To satisfy the reviewer’s demand, we also added additional references (PMID: 8889039 & 10991836), more descriptive about the ECG performance on rats (lines 105, 128-129).
[POINT 4] lines 131-132: About Bazett's calculation -> Was it automatic or not? And if it was performed by any observer what about intra- and inter observer variability?
[RESPONSE 4] The Bazett’s calculation was manually and blindly performed by one operator. This detail is now specified in the manuscript (lines 131-132). The intra- and inter observer variability was not assessed in this study because the performing operator was previously well trained to perform precisely this calculation.
[POINT 5] ABOUT REFERENCES. Please re-check if all are in accordance with authors' guidelines (i.e. number 55 --> Free Radical Biology and Medicine is a wrong abbreviation)
[RESPONSE 5] We checked and corrected the abbreviations in the references section.
Reviewer 2 Report
Dallons and colleagues aim to identify potential biomarkers for cardiotoxicity after administration of drugs that induce heart necrosis (isoproterenol), and prolongation of QT interval (clarithromycin). This was done by collecting urine samples before and repeatedly during exposure of the drugs from rats. Potential metabolic biomarkers were identified by NMR.
Firstly, I think the paper is clearly written, and the experiments and analysis seems to be properly performed. Secondly, I think this paper address an important question: That there is a need for an early identification of cardiotoxicity caused by drug administration.
Major comments:
My main concern is that the study does not necessarily answer the questions addressed in the aim. The data show an association between the urine metabolites and the use of these drugs, but this association can be caused by other biological effects. It is difficult to judge whether or not the changes in metabolic profiles in the urine is actually caused by cardiotoxicity, or whether it is caused by other effects that the drugs may have inside the body. For example: Isoproterenol may induce insulin resistance, which will most likely affect the metabolism and thus the metabolites in urine (Ref: Hoff, R; Koh, CK (2018). "Isoproterenol Induced Insulin Resistance Leading to Diabetic Ketoacidosis in Type 1 Diabetes Mellitus". Case Reports in Endocrinology. 2018). Clarithromycin has a fairly rapid first-pass metabolism in the liver, which most likely will have an effect on urine metabolites.
I still think the data may be published, but with major revisions. For example
Alternative 1: If you somehow can argue better that it is very likely that the metabolic changes (either all, or some of these) you see in the urine samples are caused by cardiotoxicity, and not anything else. I am not convinced at the moment.
Alternative 2: More clearly state that the findings are associations and not necessarily causations. Yes, there is an association between urine metabolites and these drugs, and there is a chance that some of these may be caused by cardiotoxicity. However, this is just speculations at the moment (in my opinion), and the potential biomarkers must be followed up in new studies. This means: rewrite most sections in the paper and change the title (for example: the title should not contain heart necrosis and QT prolongation. Abstract, Introduction, Discussion and Conclusion must be rewritten. The aim must be rephrased).
Alternative 3: Use the data differently. For example if you can discriminate between rats based on severity (high vs low fraction of necrosis, long vs short QT prolongation and concentrations on LDH and CPK). If you see a discrimination based on severity in animals receiving the same treatment dose, it is at least a stronger indication that the changes you see is truly caused by cardiotoxicity. The problem here may potentially be that the difference in the severity between the individual rats is not that high, and that the number of animals per group (severe vs mild symptoms) will be fairly low.
Alternative 4: Add more data: if you could include animal models (rats) that have a variable response to the same dose of treatment, and look at differences in the severity of cardiotoxicity vs urine metabolites. Alternatively, include NMR analysis of heart tissue, either extracts (regular NMR) or intact cardiac tissue (High Resolution Magic-Angle-Spinning MRS). That would more strongly indicate that the effects is truly from cardiotoxicity. The three rats receiving vehicle would be control group, and you would only have endpoint data from the two treatment group.
Minor comment:
Figure 4: It is difficult to see the difference in color (different p values) particularly between orange and red (3e-1 to 3e-3). Is it possible to display this more clearly?
Author Response
We are grateful to Reviewer 2 for his/her careful reading and comments. We propose the following answers to Reviewer’s comments:
[POINT 1] My main concern is that the study does not necessarily answer the questions addressed in the aim. The data show an association between the urine metabolites and the use of these drugs, but this association can be caused by other biological effects. It is difficult to judge whether or not the changes in metabolic profiles in the urine is actually caused by cardiotoxicity, or whether it is caused by other effects that the drugs may have inside the body. For example: Isoproterenol may induce insulin resistance, which will most likely affect the metabolism and thus the metabolites in urine (Ref: Hoff, R; Koh, CK (2018). "Isoproterenol Induced Insulin Resistance Leading to Diabetic Ketoacidosis in Type 1 Diabetes Mellitus". Case Reports in Endocrinology. 2018). Clarithromycin has a fairly rapid first-pass metabolism in the liver, which most likely will have an effect on urine metabolites.
[RESPONSE 1] We agree that the observed changes in the urine metabolic profile can be caused either by cardiac effects or other biological effects induced by the drugs. The objective of our study was to highlight potential biomarkers of two kinds of cardiotoxicity. For this purpose, we suggested, in the discussion section, links between observed metabolic changes and observed heart damages, without excluding the possibility of other causes. We did not conclude firmly that the metabolic signatures were absolutely caused by the cardiotoxicities and we suggested in the conclusion that other cardiotoxic drugs should be added in a metabonomic investigation to refine the biomarkers identification and confirm their links with cardiac damages (lines 522-524). To better emphasize this crucial point, we made several changes in the discussion section. We mentioned the isoproterenol-induced insulin resistance as a potential additional cause of increased glycosuria and other energy metabolism-related changes (lines 386-392). We mentioned other toxicities induced by clarithromycin, especially neurotoxicity and digestive system upsets, that could influence the observed metabolic signature (lines 477-483). At the end of the discussion section and in the conclusion, we also clarified the importance of a further investigation to elucidate cardiac and non-cardiac causes of the observed metabolic changes that are requested for a validation of the biomarkers (lines 502-503, 514-515, 520-521).
[POINT 2] I still think the data may be published, but with major revisions. For example
Alternative 1: If you somehow can argue better that it is very likely that the metabolic changes (either all, or some of these) you see in the urine samples are caused by cardiotoxicity, and not anything else. I am not convinced at the moment.
[RESPONSE 2] It is difficult to find more arguments to prove the link between metabolic changes and cardiotoxicity due to a limited literature, especially for certain metabolic changes we observed. We tried to support our interpretations with other studies when it was possible. For example, we mentioned the study of Li et al. (2015) reporting metabolic changes in serum induced by several cardiotoxic drugs (including isoproterenol). We noticed some similarities with our results (lines 369-371). We also mentioned 6 references suggesting a link between arrhythmia and loss of mitochondrial function (lines 444-447). To better support our interpretations regarding Krebs cycle intermediates, glucose and taurine levels changes, we added more arguments and references about the link between these changes and cardiotoxicity/cardiac damages (lines 371-375, 376-386, 413-415, 418-433). We are aware that our arguments are not so numerous, but we hope to stimulate the scientific community to investigate the complexity of metabolic changes due to cardiotoxicities.
[POINT 3] Alternative 2: More clearly state that the findings are associations and not necessarily causations. Yes, there is an association between urine metabolites and these drugs, and there is a chance that some of these may be caused by cardiotoxicity. However, this is just speculations at the moment (in my opinion), and the potential biomarkers must be followed up in new studies. This means: rewrite most sections in the paper and change the title (for example: the title should not contain heart necrosis and QT prolongation. Abstract, Introduction, Discussion and Conclusion must be rewritten. The aim must be rephrased).
[RESPONSE 3] We agree that most of our interpretations are hypotheses. We tried to construct these hypotheses with logical arguments based on previous studies found in the literature and related to this research field. We hope that these hypotheses will trigger new studies intending to deepen the question of the causality of metabolic changes and the underlying mechanisms that could be involved in cardiotoxicities. For this reason, we would like to keep the focus of this article towards cardiotoxicity and biomarkers keywords.
[POINT 4] Alternative 3: Use the data differently. For example if you can discriminate between rats based on severity (high vs low fraction of necrosis, long vs short QT prolongation and concentrations on LDH and CPK). If you see a discrimination based on severity in animals receiving the same treatment dose, it is at least a stronger indication that the changes you see is truly caused by cardiotoxicity. The problem here may potentially be that the difference in the severity between the individual rats is not that high, and that the number of animals per group (severe vs mild symptoms) will be fairly low.
[RESPONSE 4] The alternative 3 is not possible with our existing data set. When performing data validation of cardiac damages, the similarity between individual rats is so clear that we could not split the exposed rats into sub-groups based on the severity.
[POINT 5] Alternative 4: Add more data: if you could include animal models (rats) that have a variable response to the same dose of treatment, and look at differences in the severity of cardiotoxicity vs urine metabolites. Alternatively, include NMR analysis of heart tissue, either extracts (regular NMR) or intact cardiac tissue (High Resolution Magic-Angle-Spinning MRS). That would more strongly indicate that the effects is truly from cardiotoxicity. The three rats receiving vehicle would be control group, and you would only have endpoint data from the two treatment group.
[RESPONSE 5] The alternative 4 is very interesting but we do not have the possibility to perform additional experiments to collect new NMR data on heart tissues and generate different severity scale of heart damage. In our study, we focalized on urine samples because of their easy accessibility requiring no invasive procedure. Under preclinical requirement considerations, this point can be very interesting for the future development of cardiotoxicity assessment methods. However, it remains relevant to compare urine changes with those obtained from serum and heart tissues, as you suggested. This point should be particularly useful to validate the urine biomarkers.
[POINT 6] Figure 4: It is difficult to see the difference in color (different p values) particularly between orange and red (3e-1 to 3e-3). Is it possible to display this more clearly?
[RESPONSE 6] Unfortunately it is impossible to improve the color display. The figure was generated by the online MetaboAnalyst software and there is no possibility to change the color scale. We apologize for the inconvenience.
Round 2
Reviewer 2 Report
Thanks for considering my comments and concerns, and for adding relevant information into the discussion section.
I still think some of the sentences in the paper should be revised, to more clearly state that the findings in the study are not necessarily only caused by isoproterenol-induced heart necrosis or clarithromycin-induced QT interval prolongation. Here are few specific suggestions that I hope you can take into consideration
Title (line 2-5): I think “due to” is very assertive/insistent in this situation. Is it possible to replace “due to” with another word or to rephrase the title?
Abstract: line 21-23 (page 1): Instead of “evaluate the ability of 1H-NMR-based metabonomics to identify heart toxicity at very early stages”… Could you say something like “evaluate potential biomarkers of heart toxicity at very early stages using 1H-NMR-based metabonomics”?
Introduction: line 90-92 (page 2): “…urine metabonomic signatures of isoproterenol-induced heart necrosis and clarithromycin-induced QT interval prolongation…” I think the objective should be rephrased, because it is too assertive/insistent. Is it possible to say something like that you want to “characterize urine metabonomic signatures from rats receiving isoproterenol and clarithromycin, to identify candidate biomarkers for heart necrosis and QT-prolongation”?
Other minor comments:
Line 328 (page 11): Can you please add a reference here?
Line 468 (page 14): Misspelling in the sentence?
Author Response
We would like to acknowledge the Reviewer 2 for his/her new relevant comments. Herewith our answers to the Reviewer’s comments:
[POINT 1] Title (line 2-5): I think “due to” is very assertive/insistent in this situation. Is it possible to replace “due to” with another word or to rephrase the title?
[RESPONSE 1] We replaced “due to” by “presumably arising from” to make the title less assertive (lines 2-5, page 1).
[POINT 2] Abstract: line 21-23 (page 1): Instead of “evaluate the ability of 1H-NMR-based metabonomics to identify heart toxicity at very early stages”… Could you say something like “evaluate potential biomarkers of heart toxicity at very early stages using 1H-NMR-based metabonomics”?
[RESPONSE 2] We replaced “evaluate the ability of 1H-NMR-based metabonomics to identify heart toxicity at very early stages” by “evaluate potential biomarkers of heart toxicity at very early stages using 1H-NMR-based metabonomics” (lines 21-22, page 1).
[POINT 3] Introduction: line 90-92 (page 2): “…urine metabonomic signatures of isoproterenol-induced heart necrosis and clarithromycin-induced QT interval prolongation…” I think the objective should be rephrased, because it is too assertive/insistent. Is it possible to say something like that you want to “characterize urine metabonomic signatures from rats receiving isoproterenol and clarithromycin, to identify candidate biomarkers for heart necrosis and QT-prolongation”?
[RESPONSE 3] We replaced “The objectives of the present study were first to characterize urine metabonomic signatures of isoproterenol-induced heart necrosis and clarithromycin-induced QT interval prolongation in rats and, next, to isolate from those signatures early and potentially discriminant biomarkers of cardiotoxicity” by “The objective of the present study was to characterize urine metabonomic signatures from rats receiving isoproterenol or clarithromycin to identify candidate biomarkers for heart necrosis and QT prolongation.” (lines 90-92, page 2).
[POINT 4] Other minor comments:
Line 328 (page 11): Can you please add a reference here?
Line 468 (page 14): Misspelling in the sentence?
[RESPONSE 4] We added a reference for the sentence “Approximately 30% of drug withdrawals are due to unanticipated cardiac adverse events” (line 328, page 11). We replaced “decreased” by “decrease” (line 468, page14).